# mTORC2 confers neuroprotection and potentiates immunity during virus infection

Rahul K. Suryawanshi[1], Chandrashekhar D. Patil [1], Alex Agelidis [1,2], Raghuram Koganti [1], Joshua M. Ames[1,2], Lulia Koujah[1,2], Tejabhiram Yadavalli [1], Krishnaraju Madavaraju [1], Lisa M. Shantz [3] & Deepak Shukla [1,2✉]

Herpes simplex virus type-1 (HSV-1) causes ocular and orofacial infections. In rare cases, HSV-1 can cause encephalitis, which leads to permanent brain injuries, memory loss or even death. Host factors protect humans from viral infections by activating the immune response. However, factors that confer neuroprotection during viral encephalitis are poorly understood. Here we show that mammalian target of rapamycin complex 2 (mTORC2) is essential for the survival of experimental animals after ocular HSV-1 infection in vivo. We find the loss of mTORC2 causes systemic HSV-1 infection due to defective innate and adaptive immune responses, and increased ocular and neuronal cell death that turns lethal for the infected mice. Furthermore, we find that mTORC2 mediated cell survival channels through the inactivation of the proapoptotic factor FoxO3a. Our results demonstrate how mTORC2 potentiates host defenses against viral infections and implicate mTORC2 as a necessary factor for survival of the infected host.

[1] Department of Ophthalmology and Visual Sciences, University of Illinois at Chicago, Chicago, IL, USA. [2] Department of Microbiology and Immunology, University of Illinois at Chicago, Chicago, IL 60612, USA. [3] Department of Cellular and Molecular Physiology, Pennsylvania State University College of Medicine, Hershey, PA, USA. ✉email: dshukla@uic.edu

H erpes simplex virus-1 (HSV-1) is a double stranded DNA virus, which causes one of the most notorious viral infections known to humanity[1]. Globally, an estimated 3.7 billion people under age 50 (67%) harbor HSV-1 virions for life and contribute to high transmission rate via symptomatic and/or asymptomatic shedding of the virus[2]. More commonly, HSV-1 causes fever blisters and cold sores but it is also known to cause distressing ocular infections leading to blindness in untreated patients and, in severe cases, life threatening diseases such as viral encephalitis or meningitis which in turn, may result in permanent nervous system damage in surviving patients[1]. After primary infection, HSV-1 remains latent in neuronal tissues including the trigeminal ganglia and reactivates under stress situations. In general, about 74 viral proteins usurp the host assembly to make the host microenvironment permissive to virus replication in neurons as well as the mucosal epithelium, which forms the common site for symptomatic infection and virus shedding. During the infection, several host proteins combat HSV-1 replication by inducing innate and adaptive immune responses. Host cells also activate stress responses, including programmed cell death, which limits spread of virus[3]. Although apoptosis plays an important role in antiviral defense mechanisms[4], excessive cell death in neurons during neuronal HSV-1 infection may be detrimental to host survival[5]. This phenomenon might be one of the reasons behind severities during viral encephalitis. Exploring the role of pro-survival host factors including key signaling mechanisms may shed light on new ways to combat severe HSV-1 infections.

Cellular signaling through the mechanistic target of rapamycin (mTOR) plays an important regulatory role in growth, proliferation, and survival of mammalian cells[6]. It also constitutes a fundamental pathway that is essential for proper development of the brain as well as the immune system[7,8]. mTOR is a serine/threonine kinase, which belongs to the phosphatidylinositol 3-kinase (PI3K)-related protein kinase family. It forms the major catalytic subunit of two functionally distinct complexes, mTORC1 and mTORC2[6]. The mTORC1 complex is composed of mTOR, Raptor, mLST8, PRAS40, and DEPTOR[9]. This complex regulates mRNA translation and ultimately cell growth and proliferation via phosphorylation of S6K and 4EBP1. Given its significance in protein synthesis, mTORC1 activity is also known to play an important role in viral replication[10–14]. In contrast, relatively less is known about the mTORC2 complex in viral lifecycle, which consists of mTOR and Rictor (instead of Raptor) as essential components. This complex is required for the phosphorylation of AKT at $Ser^{473}$, which is in turn required for maximal AKT activation[15]. AKT plays an important role during HSV-1 infection as its inhibition impedes HSV-1 replication[16]. In addition, mTORC2 phosphorylates protein kinase C (PKC) as well as the serum and glucocorticoid induced kinases, and thus it regulates many key aspects of cytoskeleton organization, immune cell functions and cell survival[6].

Our in vivo study highlights the importance of functional mTORC2 in preventing lethality that might occur during the initial (primary) infection with HSV-1 when adaptive immunity does not exist against the virus, which is capable of infecting virtually all human cell types[17]. We demonstrate that in absence of Rictor, which is an essential component of mTORC2, HSV-1 infection shows more apoptotic cell death in corneal and neuronal tissues. mTORC2 mediated cell survival transduces through nuclear preclusion of Forkhead transcription factor FoxO3a. We also show that mTORC2 is required for maximal activation of AKT, a factor that directs inactivation of Forkhead transcription factor FoxO3a to drive pro-survival pathways during HSV-1 infection. Collectively, using a highly prevalent viral infection as a model, this study assigns a new significance to mTORC2 signaling and exposes uncharted avenues for future therapeutic interventions against viral diseases.

## Results

**Loss of Rictor causes systemic and lethal HSV-1 infection.** To determine whether the expression of Rictor, an essential component of mTORC2, is modified during HSV-1 infection, we infected human corneal epithelial (HCE) cells at varying MOIs of HSV-1. By 24 h of post-infection (hpi), Rictor transcripts increase with respect to HSV-1 infection (Supplementary Fig. 1a). However, Rictor protein levels briefly decline upon the onset of early viral gene expression such as ICP-0 and ICP-4 (Supplementary Fig. 1b). We hypothesized that there may exist an interplay between HSV-1 and Rictor, but there are no pharmacological inhibitors, which selectively target mTORC2. Furthermore, genetic deletion of Rictor is embryonically lethal[18]. To bypass these limitations, we utilized a CreERT2 model to engineer Rictor conditional knockout mice (Supplementary Fig. 1c)[19]. Upon tamoxifen treatment, Rictor can be deleted from these mice, which henceforth referred to as iRic−/− (inducible Rictor knockout) mice.

After five days of tamoxifen treatment, iRic+/+ and −/− mice were infected with HSV-1 McKrae strain at $1 \times 10^5$ MOI to their corneas (Fig. 1a). Strikingly, viral infection in the iRic−/− mice is nearly lethal. While none of the iRic+/+ mice ceased at 13 dpi, 80% of the iRic−/− mice died by the same time point (Fig. 1b). We hypothesized that a systemic infection beyond the cornea led to the animal death. We used another set of iRic+/+ and −/− mice to investigate the progress of the HSV-1 infection. Eyewashes were taken at 4 dpi (Fig. 1a). By 4 dpi, iRic−/− contained significantly more virus compared to iRic+/+ (Fig. 1c). iRic−/− mice also displayed more severe corneal damage and a greater infection score (Fig. 1d, e). In addition, iRic−/− mice lost significantly more weight than iRic+/+ and exhibited greater quantities of viral transcripts in their blood (Fig. 1f, g). These results constitute evidence of a more severe, systemic infection in the iRic−/− mice.

During ocular infection, HSV-1 travels from the cornea to the trigeminal ganglion (TG)[20]. qRTPCR of the TG tissue revealed an increased presence of the HSV-1 Lat transcripts, a marker for viral latency in the knockout mice (Fig. 1h)[21]. As the TG is a well-known site for viral latency, we further took account of latent HSV-1 in there. The ex vivo culture of murine TGs from infected iRic−/− mice, showed more viral plaques than the TG from iRic+/+ mice (Fig. 1i). Since the TG yielded more virus in the iRic−/− mice, we wanted to investigate the viral load in the brain stem, which is the next destination of HSV-1 after it reaches the TG. Both immunostaining and plaque assay results demonstrate an increased viral presence in the brain stems of the knockout mice (Fig. 1j, k). Collectively, our data indicate that the iRic−/− mice displayed a worse HSV-1 infection in the tissues of the cornea, TG, and brain stem. Thus, HSV-1 infection in the Rictor knockout mice results in a severe, systemic infection, which culminates in their death.

**Rictor deficiency results in decreased innate and adaptive immune responses.** Given the disparities in survivability between the iRic+/+ and −/− mice during infection, we wanted to identify whether the immune response of the knockout mice was intact. Murine corneas taken at 10 dpi exhibit greater immune cell infiltration and inflammation in the wild-type mice (Fig. 2a, b). We then checked the presence of a panel of immune cells using flow cytometry on the whole eye tissue. Here too, the iRic+/+ possessed greater numbers of helper T cells (Fig. 2c), cytotoxic T cells (Fig. 2d), active T cells (Fig. 2e), dendritic cells (Fig. 2f), and NK cells (Fig. 2h). The populations of plasmacytoid dendritic cells remained similar across the two groups (Fig. 2g). We confirmed the influx of CD8a T cells in the wild-type corneal tissue via IHC (Fig. 2i). In contrast basal level of immune cells

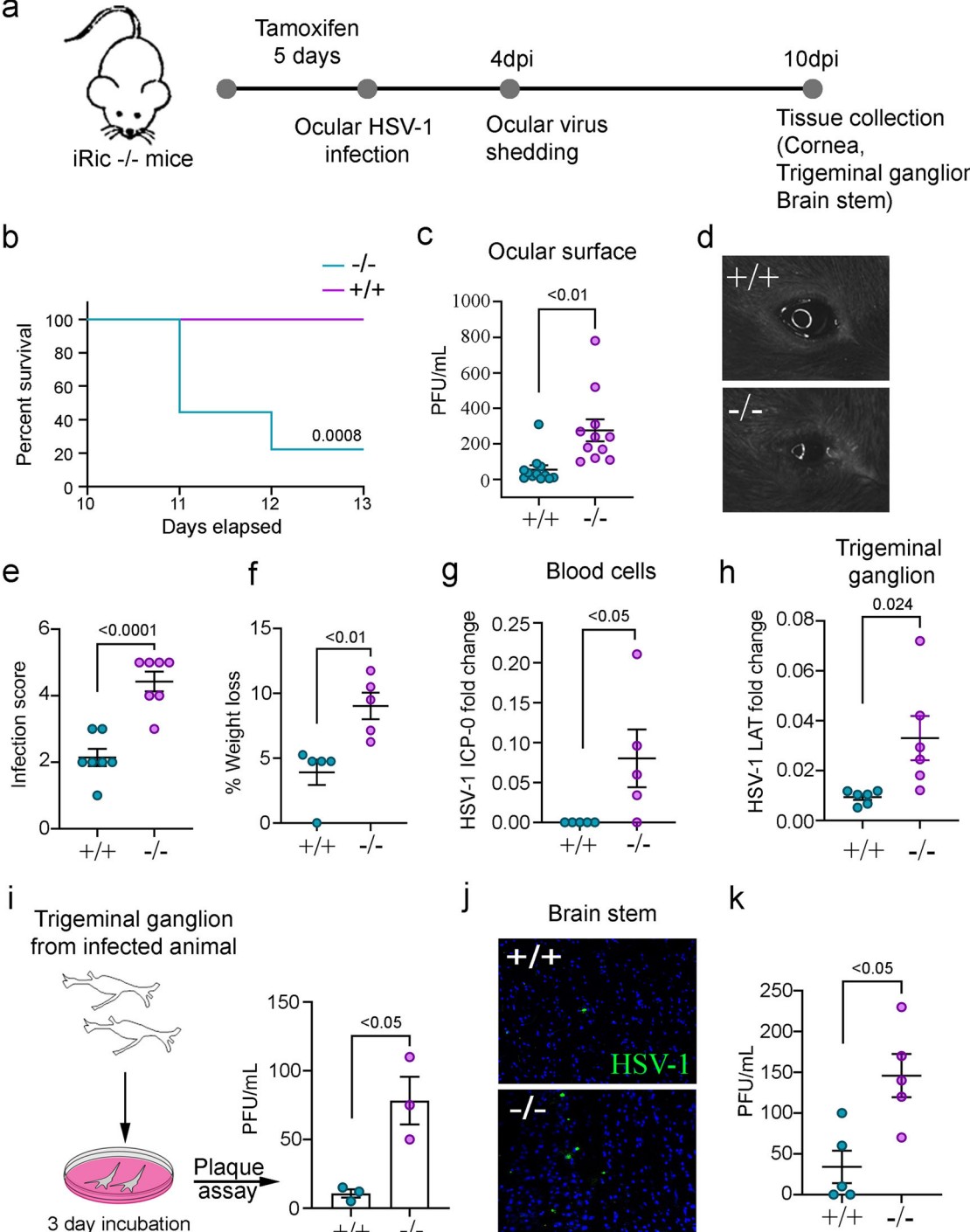

**Fig. 1 Lack of Rictor is lethal and causes systemic HSV-1 infection. a** Schematics of the experiment showing conditional knockout of the iRic−/− mice using Tamoxifen or mock. The mice were infected with HSV-1 McKrae and analyzed for systemic infection. **b** Survival curve showing percent survival after iRic+/+ and −/− infection ($n = 10$). Log-rank (Mantel-core) test was used to analyze the survival curve data. **c** Plaque assay showing mature virus particle formation in eyewash samples collected at 4 dpi ($n = 11$). **d** Representative eye image of HSV-1 infected animals at 4 dpi. **e** Graph showing image score based on visual observation at 6 dpi ($n = 7$). **f** Graph representing percent weight loss at 12 dpi ($n = 5$). **g** qRTPCR analysis of early viral gene isolated from blood cells at 5 dpi ($n = 5$). **h** qRTPCR analysis of late viral gene isolated from trigeminal ganglion (TG) cells at 5 dpi ($n = 6$). **i** To reactivate the virus in TG of infected mice, TG were cultured in vivo and the mature virus particles were analyzed with plaque assay ($n = 3$). **j** A representative micrograph of immunohistochemistry analysis for HSV-1 detection in brain stem at 10 dpi. **k** Graph representing mature virus particle formation in brain stem at 5 dpi ($n = 5$). Two-tailed unpaired $t$-test was used to analyze the data presented in **c**, **e–i**, and **k**. Data are represented as mean ± SEM in **c**, **e–i**, and **k**. Data in the micrograph (**d**, **j**) is representative of five independent experiments. Source data underlying Fig. 1b are provided as a Source Data file.

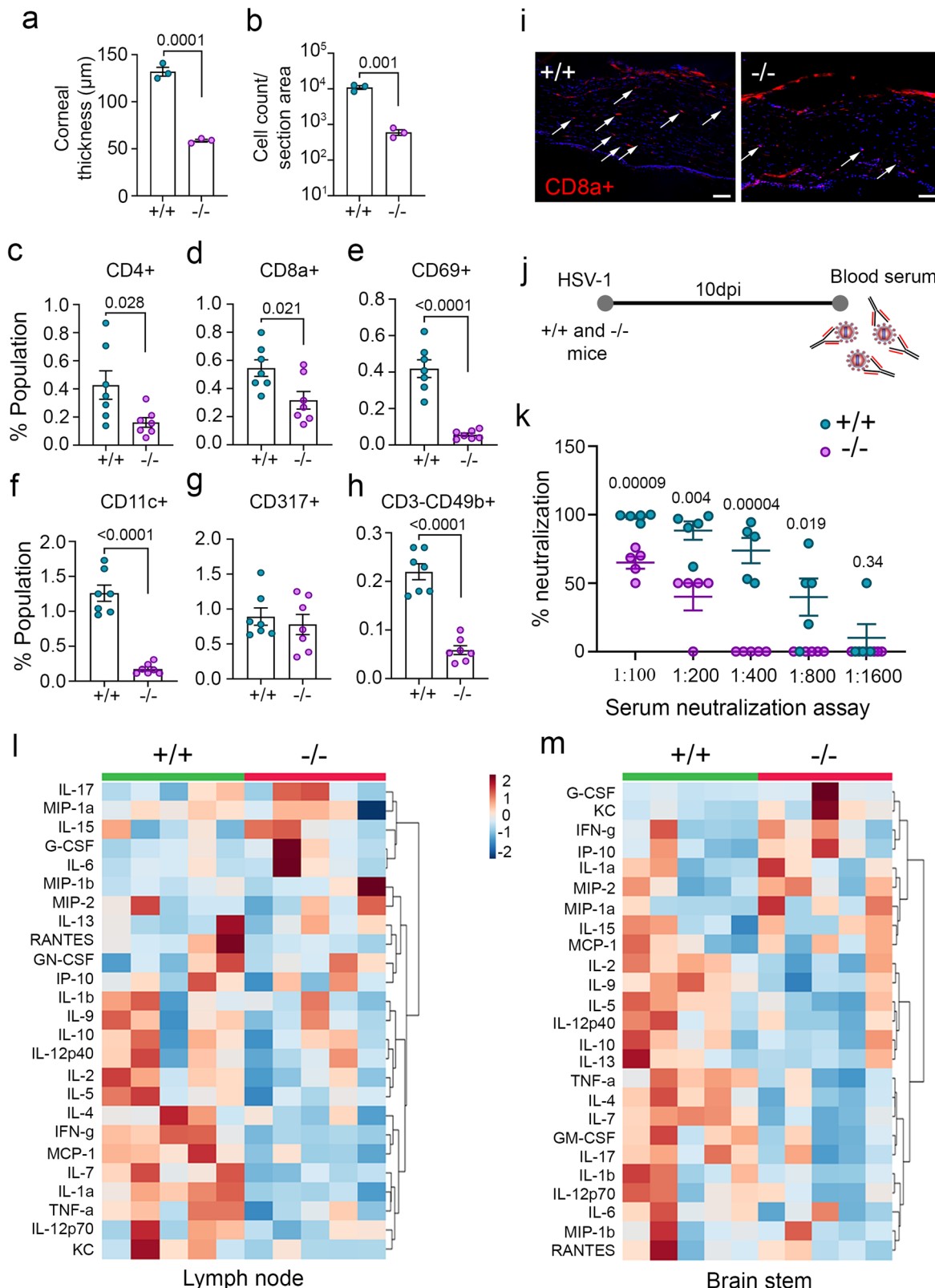

were similar in mock infected animals except for plasmacytoid dendritic cells, which found to be low in iRic−/− animals (Supplementary Fig. 2).

Since the T cell populations were decreased in the infected iRic−/− mice, we measured whether the B cell response was attenuated as well. After infecting both groups of mice and isolating blood serum at 10 dpi, we performed a serum neutralization assay

(Fig. 2j). The concentration of neutralizing antibodies was far greater in the wild-type mice than the knockouts, suggesting that both humoral and cell-mediated immunity were defective in the iRic−/− mice (Fig. 2k).

As both arms of the adaptive immune response were attenuated in the Rictor knockout mice, it was deemed possible that their activation and coordination were impaired. Using tissue

**Fig. 2 Rictor deficiency results in defective innate and adaptive immune responses. a** A graph representing corneal thickness of HSV-1 infected mice, quantified from H and E stained tissue sections, ($n = 3$). **b** Graph representing the cell count per section area for HSV-1 infected cornea of mice, quantified from H and E stained tissue sections ($n = 3$). **c–h** Graph representing population of respective immune cells in HSV-1 infected eye at 10 dpi, ($n = 7$). **i** Immunohistochemistry staining of CD8+ cells in HSV-1 infected corneal tissue. Data is the representative of five independent experiments. Scale bar-50 μm. **j** Schematics of neutralization assay. **k** Graph showing neutralizing antibody response from serum collected at 10 dpi ($n = 5$ per group). Multiple $t$-test was performed assuming individual variance for each row. **l** Cytokine analysis of lymph node and **m** brain stem tissue of HSV-1 infected animals at 10 dpi ($n = 5$). Two-tailed unpaired $t$-test was used to analyze the data presented in **a–h**. Data are represented as mean ± SEM in **a–h** and **k**. Source data underlying Fig. 2k–m are provided as a Source Data file.

isolated from cervical lymph nodes and the brain stem, we explored the cytokine profile of each group of mice using a Luminex cytokine panel. In both the tissues, the iRic−/− mice showed a weakened cytokine response compared to their wild-type counterparts (Fig. 2l, m). Furthermore, the types of cytokines activated by each group of mice significantly differed. In the brain stem, the pro-inflammatory cytokines IL-1α, IL-1β, and TNF-α were up-regulated in the wild-type mice (Supplementary Fig. 3). However, the expression of the anti-inflammatory cytokines IL-2, IL-10, IL-13, IL-17 was increased in the Rictor knockout mice, both at baseline and after infection (Supplementary Fig. 3), the results are in line with previous reports[22,23].

**Rictor is an important cell survival factor during virus infection.** mTORC2 plays important roles in balancing cell survival and apoptosis to promote homeostasis of the cell[24]. Since their immune function is significantly compromised during HSV-1 infection (Fig. 2), we hypothesized that iRic−/− mice may be more likely to undergo cellular apoptosis to mitigate viral spread. This shift in homeostasis towards cell death may account for the difference in survivability between the iRic +/+ and −/− mice. To generate evidence for our hypothesis, we performed TUNEL staining of the corneas of infected mice at 4 dpi and observed that the knockout mice show a marked pattern of cell death along the cornea and retina, which extends to the optic nerve (Fig. 3a). In parallel, TUNEL staining of the brain stem at 10 dpi highlighted an increase in cell death in the iRic−/− mice (Fig. 3b). However, iRic+/+ showed relatively little cell death in these two tissues. The cell death in ocular tissue and neurons may be attributed to impaired AKT activation by mTORC2[25,26].

To study the apoptotic responses of iRic−/− mice in vitro, we isolated mouse embryonic fibroblasts (MEFs) from murine fetuses and treated them with 4-hydroxytamoxifen (Fig. 3c). Thus, we were able to culture iRic+/+ and −/− cells for in vitro experiments. In agreement with our in vivo results, we found that the iRic−/− MEFs showed increased levels of both early and late stage apoptosis (Fig. 3d). We also took note of basal level apoptosis, which was observed to be significantly higher in iRic−/− cells compared to iRic+/+ cells (Supplementary Fig. 4e), and the level of cell death as estimated by PI staining increases with HSV-1 infection or etoposide treatment (Supplementary Fig. 4). Pertaining to the evidence of apoptosis, caspase 3/7 activation was found to be enhanced in the iRic−/− MEFs (Fig. 3e).

Next, to understand the factors contributing to the onset of apoptosis in iRic−/− cells, we analyzed relative levels of proteins in an apoptosis protein array (Fig. 3f, g). Interestingly, an anti-apoptotic protein, myeloid leukemia cell differentiation protein 1 (MCL-1), was preferentially up-regulated in iRic+/+ cells whereas several pro-apoptotic proteins including tumor necrosis factor receptor 1, hypoxia inducing factor and HSP-27 were upregulated in infected iRic−/− cells. If loss of Rictor results in an increase in cell death upon viral infection both in vitro and in vivo, then an inhibitor of apoptosis may promote host cell survival and prevent the lethality of infection in iRic−/− mice. We therefore used a pan-caspase inhibitor Z-VAD-FMK to

pharmacologically reduce apoptosis in both the iRic−/− MEFs and mice (Fig. 3g). Z-VAD-FMK treatment resulted in a loss of cell death in the MEFs (Fig. 3h). Critically, Z-VAD-FMK rescued the iRic−/− mice as 60% of the mice survived to 15 dpi as opposed to only 20% in the mock-treated group (Fig. 3i). The results imply that the Z-VAD-FMK mediated survival of HSV-1 infected iRic−/− animals is not attributable to the increase in immunity (Supplementary Fig. 5a–g).

Apoptosis is an important mechanism to restrict virus spread, and the absence of Rictor exacerbates apoptosis. We further aimed to analyze HSV-1 replication in iRic−/− cells, and, as expected due to their susceptibility for apoptosis, iRic−/− cells showed restricted HSV-1 replication compared to iRic+/+ cells. The results were evident through fluorescence imaging, flow cytometry of infected cells, western blot and plaque assay (Supplementary Fig. 6a–d). We also confirmed that the virus restriction in iRic−/− cells is not because of cytokine production as the iRic−/− cells showed significantly low expression of cytokine upon infection (0.1 or 1 MOI) with HSV-1 KOS (Supplementary Fig. 7). The increase in virus replication in iRic−/− mice must be attributed to immune dysfunction (Fig. 2), including dysregulated secretion of inflammatory and regulatory cytokines. The results highlight the importance of both cell death and immune response in regulating the virus infection, and underline the significance of mTORC2 during the virus infection.

**mTORC2 mediates cell survival through AKT mediated inactivation of FoxO3a.** After observing the extensive cell death responses in iRic−/− MEFs, we wanted to elucidate the mechanism of the observed apoptosis. mTORC2 has been shown to phosphorylate AKT at Ser-473 which activates it[27]. AKT then proceeds to phosphorylate the pro-apoptotic transcription factor forkhead box O3a (FoxO3a)[28] at Ser-253 and Thr-32. Once phosphorylated, FoxO3a becomes inactive and it is precluded from the nucleus, which in turn, prevents it from stimulating apoptosis (Fig. 4a)[28]. A western blot time course of infected HCE cells revealed that phosphorylation of AKT at 8 hpi is closely followed by phosphorylation of FoxO3a at 10 hpi (Fig. 4b and Supplementary Fig. 8). A similar phenomenon occurs in Lund human mesencephalic (LUHMES) neuronal cells whereby both AKT and FoxO3a become phosphorylated by 24 hpi, highlighting the mTORC2-Akt-FoxO3a axis as a conserved response during HSV-1 infection (Fig. 4c). We hypothesized that loss of Rictor would inhibit AKT phosphorylation, which would allow FoxO3a to facilitate the transcription of pro-apoptotic factors. We found that deletion of Rictor abrogated phosphorylation of both AKT and FoxO3a during virus infection (Fig. 4d). In contrast, the phosphorylation of both proteins occurred in a dose-dependent manner with infection in iRic+/+ MEFs (Fig. 4d). If the phosphorylation of FoxO3a is inhibited, it should be present inside the nucleus and vice versa. Using confocal microscopy, we found that FoxO3a was precluded from the nucleus during HSV-1 infection in iRic+/+ MEFs (Fig. 4e). However, FoxO3a appeared inside the nucleus in the iRic−/− MEFs, localizing within the DAPI stain (Fig. 4e). Thus, loss of Rictor prevents the downstream

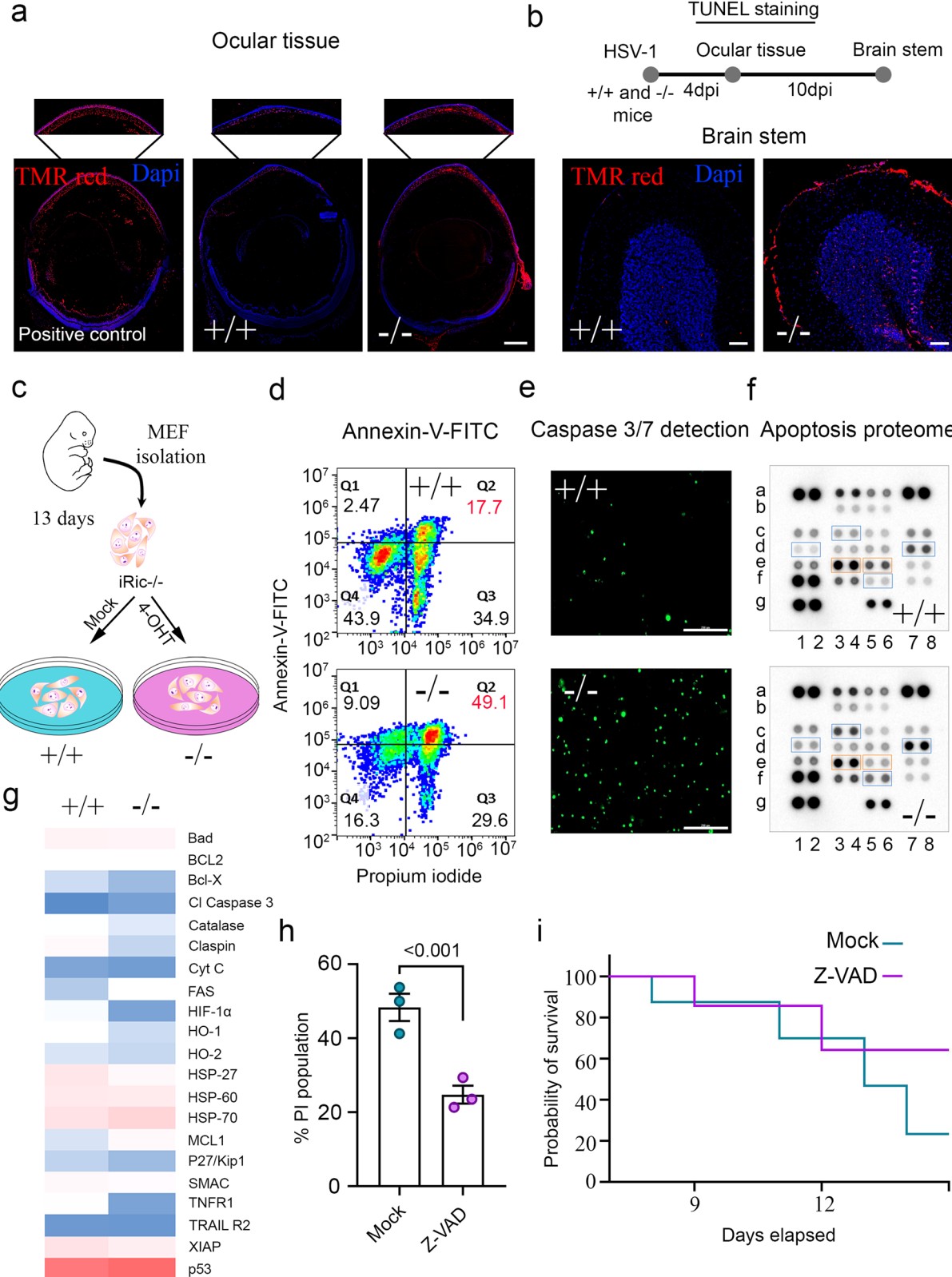

phosphorylation of AKT and FoxO3a, which stimulates cellular apoptosis.

Finally, we hypothesized that loss of FoxO3a reduces the cell's ability to mitigate infection via apoptosis. By using cells with a genetic deletion of FoxO3a, we observed an increase in virus infection in the FoxO3−/− cells as compared to the FoxO3a+/+ cells (Fig. 4f–h). Consistent with this result, FoxO3a−/− cells

demonstrate slower apoptosis kinetics during infection (Fig. 4i, j). iRic−/− cells conversely show an increased rate of cell death during HSV-1 infection or treatment with the agonist etoposide, a pharmacological promoter of apoptosis (Supplementary Fig. 5a–d). The results are in congruent with our hypothesis that during HSV-1 infection nuclear preclusion of FOXO3a mediated through mTORC2-AKT axis is prerequisite for cell survival.

**Fig. 3 Absence of Rictor induces apoptosis. a** A representative micrograph of TUNEL stained whole eyeball tissue at 4 dpi. Scale bar-500 μm. **b** Schematic and micrograph showing TUNEL staining of brain stem at 10 dpi, ($n = 5$). Scale bar-100. **c** Schematic of mouse embryonic fibroblasts (MEFs) isolation showing deletion of Rictor by 4-hydroxitamoxifen treatment. **d** Annexin-V-FITC and PI staining of MEFs at 12 h of post-infection (hpi). Q1. Annexin V+/PI− cells indicate cells undergoing early apoptosis, Q2. Annexin V+/PI+ indicate late apoptotic cells, Q3. Annexin V−/PI+ indicate mechanically damaged cells, and Q4. Annexin V−/PI− indicate living cells. **e** A representative micrograph of caspase 3/7 activity using CellEvent™ Green fluorescence indicate active caspase 3/7, Scale bar 200 μm. **f** Representative micrograph of apoptotic array proteins. **g** A heat map showing the quantitative analysis of proteome profiler mouse apoptosis array proteins from **f**. **h** A graph showing percent PI stained population of Z-VAD-FMK (10 μM) treated and HSV-1 infected iRic KO MEFs ($n = 3$). Two-tailed unpaired $t$-test was used to analyze the data. Data are represented as mean ± SEM. **i** A Survival graph showing probability of survival for Z-VAD-FMK (5 mg/kg) or mock treated HSV-1 infected iRic−/− animals ($n = 7$). Data in the micrograph (**a**, **b**, **e**) is the representative of five independent experiments. Source data underlying Fig. 3f, g are provided as a Source Data file.

## Discussion

In this study we have highlighted mTORC2 as an essential host component that is required to thwart potentially lethal consequences of a primary HSV-1 infection. We demonstrate that inactivation of mTORC2 activity causes lethality in mice upon HSV-1 infection. The iRic−/− animals show decreased immunity and increased cell death in corneal and neuronal tissues making it more evident that an active mTORC2 contributes to survival of HCE and neuronal cells by prohibiting apoptotic cell death. We attribute this function to mTORC2-mediated activation of AKT, and subsequent downregulation of FoxO3a.

The iRic−/− mice when compared to the wild type animals displayed exacerbated infection and damage in tissues such as the cornea, TG, and the brainstem, which are considered key tissues targeted by HSV-1 in humans[29]. Our results also demonstrate that HSV-1 causes a systemic infection in Rictor knockout mice compared to a more localized infection in wild type animals. The former is evidently due to malfunctioning of the immune system. Our results indicate that both arms of the immune response are attenuated in the Rictor knockout mice. We examined cytokine profile of tissue isolated from cervical lymph nodes and the brainstem of each group of mice using a Luminex cytokine panel. The unique patterns observed for the two groups suggest that mTORC2 may limit the expression of anti-inflammatory cytokines highlighting important role of mTORC2 in their homeostasis. The iRic−/− mice appear to have an intrinsically weakened innate immune system which cannot adequately stimulate adaptive immune responses during a primary HSV-1 infection. In agreement with our results, mTORC2 has been reported to be required for the induction of CCL5 in neurons and astrocytes during infection[5]. Collectively, these findings indicate that iRic−/− mice have deficiencies in the activation and proper functioning of the adaptive immune response. Related to this is the fact that we observed an impaired B cell response in absence of mTORC2, which might be due to multiple reasons including downregulation of IL-7 receptor and functional follicular helper CD4+ T cell that results in IgM+ immature B cell generation[10,11]. It is known that mTORC2 plays an important roles in balancing cell survival and apoptosis to promote cellular homeostasis[14]. We also found evidence that the iRic−/− mice are unable to mount stronger immune response to invading HSV-1, resort to undergo cellular apoptosis to mitigate a fast spread of invading as well as replicating virions. This shift in homeostasis towards cell death including the death of key neurons may eventually account for the difference in survivability between the iRic+/+ and −/− mice. Increased cell death in the eye as well as the nervous system tissues could be attributed, at least in part, to impaired AKT activation by mTORC2[15,16]. A dynamic balance between apoptosis related proteins is known to regulate apoptosis. Interestingly, an anti-apoptotic protein, MCL-1 was strongly upregulated in the iRic+/+ cells. MCL-1 is known to attenuate apoptosis and promote survival of neuronal cells[17]. Reduced levels of MCL-1 in the iRic−/− MEFs verifies the importance of

Rictor in stabilizing the protein[18]. Suppression of MCL-1 also correlates with upregulation of PUMA, which is under transcriptional regulation of proapoptotic protein FOXO3a[19]. The cumulative evidence suggests that mTORC2 acts to inhibit the apoptotic response.

Collectively, our findings suggest that during HSV-1 infection, loss of Rictor prevents the mTORC2 complex from phosphorylating AKT. Inactive AKT cannot phosphorylate FoxO3a which allows it to stimulate apoptosis. Mice with infected brain stems experience an increase in neuronal cell death, which becomes lethal over time. In light of this information, we propose that mTORC2 is essential to control HSV-1 infection and cell survival during stress situation (1) by inducing innate and adaptive immune responses and (2) protecting host cells and neurons from stress-induced apoptosis. During HSV encephalitis, neuronal apoptosis may contribute to severe symptoms such as memory loss, facial, or partial paralysis, and even death in some individuals. Acyclovir treatment is usually given well after the virus has reached the brain. Alongside of acyclovir, as we show, apoptosis inhibitor like Z-VAD-FMK may support mTORC2's functions in host survival and serve as an adjuvant therapy for severe HSV-1 infections, reducing neurodegeneration during encephalitis, however further studies are essential. Furthermore, potentiating the pro-survival and immune boosting activities of mTORC2 may comprise a possible treatment for severe cases of viral infections in general.

## Methods

**Cells and viruses**. Human corneal epithelial (HCE) cell line (RCB1834 HCE-T) was procured from Kozaburo Hayashi (National Eye Institute, Bethesda, MD). HCE's were cultured in minimum essential medium (MEM) (Life Technologies, Carlsbad, CA) supplemented with 10% fetal bovine serum (FBS) (Life Technologies) and 1% penicillin/streptomycin (Life Technologies).

Mouse embryonic fibroblasts (MEFs) with inducible Rictor−/− were isolated from a mouse having Rictor conditional alleles able to transiently express tamoxifen-inducible Cre recombinase (CreERT2). Rictor knockout in the cells was achieved by treating them with 4-hydroxytamoxifen (4OHT) (2 μM) for 72 h. MEFs with FoxO4+/−, p53−/− immortalized FoxO3a+/+ and FoxO3a−/− were a generous gift from Prof. Nissim Hay (University of Illinois at Chicago, Chicago, IL). DMEM containing 10% FBS and 1% penicillin/streptomycin was used to grow all MEFs. LUHMES cells (ATCC- CRL 2927) were provided by Dr. David Bloom (University of Florida, Gainesville, FL). HCE cells were provided by Dr. Kozaburo Hayashi (National Eye Institute, Bethesda, MD). VERO cells were a generous gift from Prof. Patricia Spear, Northwestern University, Chicago. Two different strains of HSV-1 used in study were KOS-WT (provided by Prof. Patricia Spear, Northwestern University, Chicago, IL) and McKrae (provided by Dr. Homayon Ghiasi, Cedars-Sinai Medical Center, Los Angeles, CA). The dual color ICP0p-GFP/gCp-RFP was gifted by Dr. Paul Kinchington (University of Pittsburgh, Pittsburgh, PA). HSV-1 17 strain was obtained from Dr. Richard Thompson (University of Cincinnati, Cincinnati, OH). The virus stocks were made in Vero cells and stored at −80 °C.

**Inducible Rictor knock out animal generation**. Rictor^F/F mutant mice possessing loxP sites flanking exon 11 of the RPTOR independent companion of MTOR, complex 2 (Rictor) gene (STOCK Rictortm1.1Klg/SjmJ) and whole body cre/ERT2 mice (B6.129-Gt(ROSA)26Sortm1(cre/ERT2)Tyj/J) were purchased from The Jackson Laboratory. The mice were bred to create a inducible Rictor knockout mice

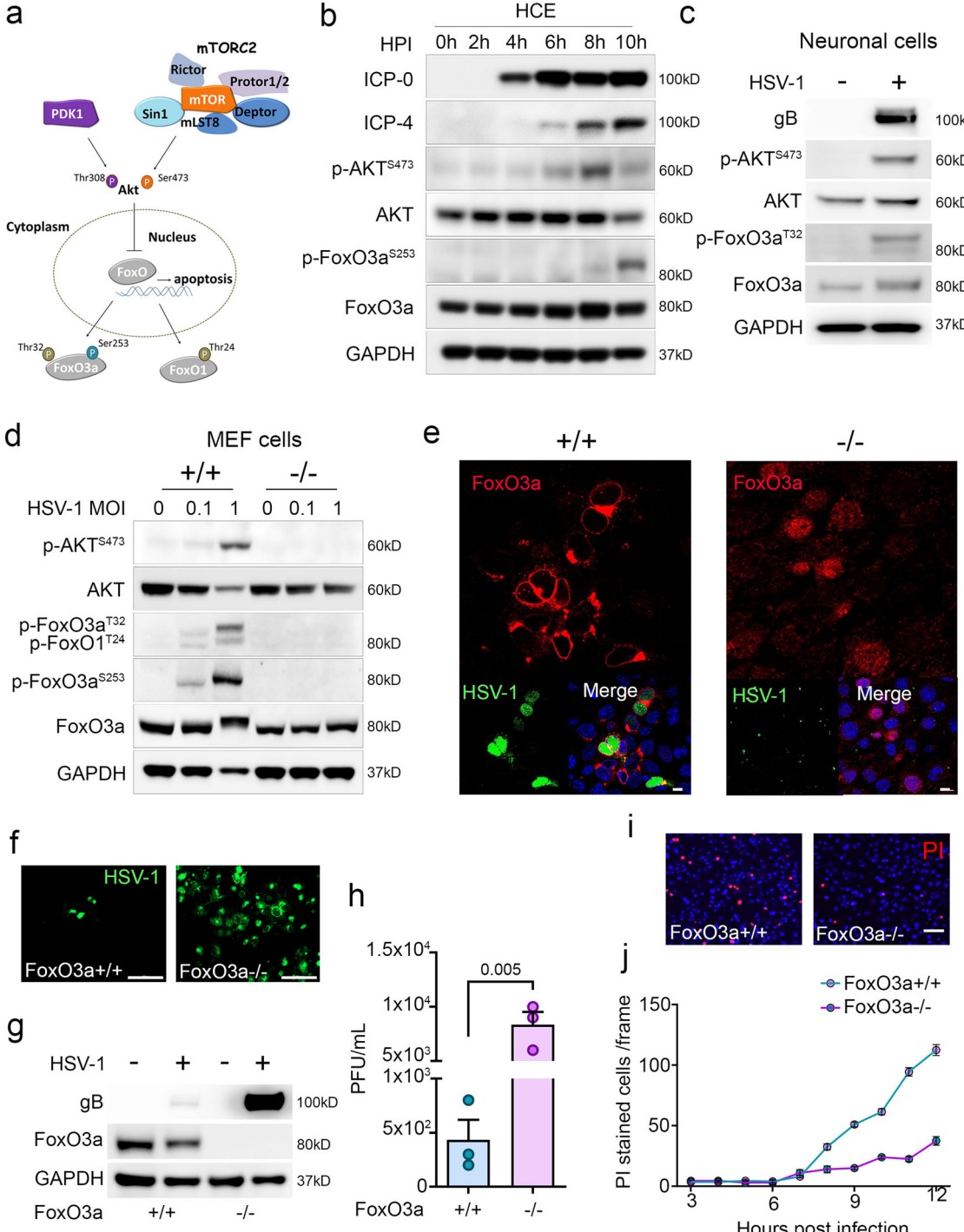

**Fig. 4 Rictor/mTORC2 dependent nuclear preclusion of FOXO3a is important for cell survival during virus infection. a** Schematic representation of the hypothesis. **b** Representative western blots showing expression and phosphorylation of respective proteins at different hpi. **c** Representative western blot illustrating expression and phosphorylation of respective proteins in LUHMES cells. **d** Representative western blot illustrating expression and phosphorylation of respective proteins in iRic MEF's. **e** Representative micrograph of immunofluorescence confocal imaging illustrating location of FoxO3a, Scale bar 10 µm. **f** Representative micrograph of fluorescence imaging in HSV-1 (GFP tagged) infected cells, Scale bar 200 µm. **g** Representative micrograph of western blot showing indicated protein expression. **h** Plaque assay showing PFU/mL (*n* = 3), two-tailed unpaired *t*-test was used to analyze the data. **i** A representative micrograph showing PI stained population of HSV-1 infected MEFs at 12 hpi, Scale bar 50 µm. **j** A line graph showing PI stained population of HSV-1 infected MEFs. Data are represented as mean ± SEM in **h**–**k**. Data in the micrograph (**b**–**g**, **i**) is the representative of three independent experiments. Source data underlying b–d, g are provided as a Source Data file.

(iRic−/−). In order to knock out Rictor from iRic−/− animals, they were treated with tamoxifen (2 mg/kg) for five days and then used for the experiment.

**Antibodies**. For Western blotting and immunofluorescence imaging: HSV-1 viral proteins were detected by ICP0 (ab6513) 1:1000, gB (ab6506) 1:1000 both purchased from Abcam (Cambridge, United Kingdom. Following antibodies were used for western blot or immunofluorescence were FoxO3a (2497S) 1:1000, AKT (9272S) 1:1000, p-FoxO3a$^{S253}$ (9466S) 1:500, Rictor (2114S) 1:1000, p-FoxO1$^{T24}$/p-FoxO3A$^{T32}$ (9464S) 1:500, p-AKT$^{S473}$ (9271S) 1:500, FoxO1 (2880) 1:1000, IRF7 (4920) 1:1000 all purchased from cell signaling technology (Danvers, MA), GAPDH (10494-1-AP) 1:2000 was purchased from Proteintech Group, Inc., (Rosemont, IL).

For flow cytometry: Flow antibodies were purchased from Biolegend were CD3 (100236), CD69 (104507), CD11b (101206), CD11c (117310), CD49b (108907), CD317 (127104) and those purchased from Tondo biosciences, San Diego were CD4 (50-0042-U100) and CD8a (20-18886-U100).

**Chemical reagents**. Pharmacological inhibitors including 4-hydroxytamoxifen (S7817), Z-VAD-FMK (S7023), Etoposide (S1225) were purchased from Selleckchem (Houston, TX).

**Western blot**. The cells in a reaction well were collected using Hanks Cell Dissociation buffer. Protein lysates were prepared using radioimmunoprecipitation assay (RIPA) buffer (Sigma Aldrich, St Louis, MO) as per manufacturer's guidelines. The protein samples were mixed with LDS sample loading buffer at 4× concentration followed by addition of beta-mercaptoethanol (5%) (Bio-Rad, Hercules, CA). The resultant mixture was denatured at 95 °C for 9 min. The protein samples were then electrophoresed on Invitrogen™ Mini Gel Tank (Fisher scientific) through pre-casted gels (4–12%). The proteins were blotted on nitrocellulose membrane (Fisher scientific). The membrane was blocked in 5% milk/TBS-T for 1 h followed by overnight incubation with primary antibody at 4 °C. The membranes were washed with TBS-T and incubated with respective horseradish peroxidase (HRP) conjugated secondary antibody (anti-mouse 1:10000 or anti-Rabbit 1:10,000) for 1 h at room temperature. 1:2500 concentrations of anti-rabbit were used for detection of all phosphoproteins. The membranes were washed again before exposing them to SuperSignal West Femto maximum sensitivity substrate (Thermo Scientific, Waltham, MA) and proteins were visualized using the with Image-Quant LAS 4000 biomolecular imager (GE Healthcare Life Sciences, Pittsburgh, PA). Full scans for the blots have been presented in source data.

**Quantitative real time polymerase chain reaction**. RNA isolation was performed using TRIzol (Life Technologies). High Capacity cDNA Reverse Transcription kit (Life Technologies) was used to transcribe RNA to cDNA using High Capacity RNA–to-cDNA Kit (Applied Biosystems Foster City, CA). The cDNA was further used for Real-time quantitative polymerase chain reaction (qPCR) performed on QuantStudio 7 Flex system (Invitrogen™ Life Technologies) using Fast SYBR Green Master Mix (Life Technologies). The primer sequences used to determine the gene expression are listed in Supplementary Table 1.

**Plaque assay**. Either HCE or MEFs were infected with either mock or 0.1 MOI of HSV-1 and incubated at 37 °C, 5% CO$_2$. After 2 hpi the inoculum medium was replaced by whole medium (MEM or DMEM containing 10% FBS and 1% PenStrap). The cells were incubated at 37 °C, 5% CO$_2$ for 24 h unless specified. After infection the cells were collected using Hanks buffer and suspended in Opti-MEM (Thermo Fisher Scientific). The resultant mixture was sonicated to produce lysates, which were used to quantify infectious virus particles. The VERO cells were seeded in tissue culture plate to form a monolayer. The monolayer was infected with respective dilution of cell lysate or eye wash in Opti-MEM. At 2hpi the dilutions of cell lysate were aspirated and replaced by a complete medium (DMEM with 10% FBS and 1% PenStrap) containing 0.5% methylcellulose (Fisher Scientific). The cells were incubated at 37 °C, 5% CO$_2$, for 72 h. After incubation the cells were fixed by adding 100% methanol. After 20 min of fixation the medium was removed and cells were stained with crystal violet solution. Numbers of plaques were counted visually.

**Immunofluorescence microscopy**. HCE or MEFs were cultured in glass bottom dishes (MatTek Corporation, Ashland, MA). Twenty-four hpi the cells were fixed with 4% paraformaldehyde (PFA) and permeabilized with 0.1% triton X-100 for 10 min for intracellular staining with a mouse or rabbit monoclonal antibody against the target protein (1 h) followed by a secondary antibody conjugated with FITC- or Alexa Fluor 647 (Sigma-Aldrich) (1.100) for 1 h at room temperature. NucBlueLiveReady Probes Hoescht stain (Thermo R37605) was used to stain the nucleus (histone). The samples were investigated under 63× oil immersion objective using LSM 710 confocal microscope (Zeiss). The resulting images were processed using ZEN black software.

**Live cell imaging**. iRic+/+ and iRic−/− MEFs were cultured in 12 well plate and infected with HSV-1 dual color fluorescent virus expressing GFP and RFP driven by ICP0 and gC promoter respectively. At 2 hpi, inoculation media was replaced with complete DMEM, and cells were placed in the incubation chamber of Zeiss

spinning disk live-cell imaging system, which maintains 37 °C and 5% CO$_2$. Images were captured for RFP, GFP, and brightfield at an interval of 30 min for 24 h, and analyzed with ZEN software. Similar experiments were done for estimation of propidium iodide (PI) uptake for iRic+/+ and iRic−/− MEFs either infected with HSV-1 infection or treated with etoposide (10 μM). PI uptake experiment was done for HSV-1 infected FoxO3a+/+ and FoxO3a−/− MEFs as well. In all experiments NucBlueLiveReady Probes Hoescht stain was used to stain the nucleus (Thermo R37605).

**Infection to murine cornea**. All animal care and procedures were performed in accordance with the institutional and NIH guidelines, and approved by the Animal Care Committee at University of Illinois at Chicago (ACC protocol 17-077). iRic-/- C57BL/6J female mice were either dosed with mock or tamoxifen (2 mg/kg) for 5 days before infection. At the time of infection mice were anesthetized and their corneas were scarified in a 3 × 3 grid using a 30-gauge needle. The corneas were infected with HSV-1 McKrae at 1 × 10$^5$ MOI. At day 4 of post-infection eye wash samples were collected to quantify HSV-1 replication through plaque assay. Acute weight loss (>20%) and hunching of the animal was considered as a clinical end point. In a different set of experiment, mice were checked for presence of HSV-1 replication in tissues like blood, trigeminal ganglion and brain stem. The apoptosis in ocular tissue and brain stem was analyzed by TUNEL staining as per manufacturer's protocol (abcam inc.).

**Proteome profiler apoptosis array**. iRic+/+ and iRic−/− cells were infected with HSV-1 at 1 MOI. After 12 hpi the cells were rinsed with PBS and the cells were lysed as per the manufacturer's guidelines. The resultant mixture was centrifuged at 14,000 × g and the supernatant was used to estimate the total protein by using Pierce™ BCA Protein Assay Kit (Thermo scientific). Equal quantity of protein (300 μg) was loaded onto each membrane. Detection of apoptosis related protein was further performed by the manufacturer's guidelines (Proteome profiler™ array).

**Apoptosis detection by flow cytometry**. iRic+/+ and iRic−/− cells were infected with HSV-1 at 1 MOI and harvested at 8 and 12 hpi. FITC Annexin V/Dead Cell Apoptosis Kit (Invitrogen™) was used to detect apoptosis. As per the manufacturer's protocol the sample pellets were washed with cold PBS and suspended in 1× annexin binding buffer. The suspension was mixed with 5 μL FITC Annexin V and 1 μL of a propidium iodide solution (100 μg/mL) for 15 min in the cold. The mixture was diluted with 1× annexin binding buffer. Further flow cytometry (BD Accuri™ C6 Plus flow cytometer) was used to analyze the staining for Annexin V and propidium iodide. Flow data was analyzed using FlowJo software (Tree Star Inc.).

**Caspase3/7 activity assay**. CellEvent™ Caspase3/7 green detection reagent was used for analyzing the activity of caspase 3/7 in iRic+/+ and iRic−/− cells were infected with HSV-1 at 1 MOI. After 12 hpi the cells were imaged using fluorescence microscopy. Caspase 3/7 activity in cells was identified by appearance of fluorescing bright green cells.

**Flow cytometry**. In order to analyze the immune response of animals infected with HSV-1 McKrae strain, the ocular tissue was digested with collagenase and the dissociated cells were sieved with 70 μm nylon filter. The cells were washed with flow buffer and labeled with the respective flow antibodies for 1 h on ice. The unstained cells and cell stained with single color antibody were used for gating purpose. The immunolabelled cells were analyzed with Accuri C6 Plus flow cytometer (BD Biosciences). BD Accuri C6 Plus software and Tree star FlowJo v10.0.7 were used for all flow cytometry data analysis. The getting strategy have been presented in Supplementary Fig. 9.

**Statistics analysis**. Error bars of all Figures represent SEM of three independent experiments ($n = 3$), unless otherwise specified. The experimental dataset between the two groups have been compared using the two tailed unpaired Student's $t$-test or 2way ANOVA. Differences between values were considered significant when *$p < 0.05$, **$p < 0.01$, ***$p < 0.001$, ****$p < 0.0001$.

**Reporting summary**. Further information on experimental design is available in the Nature Research Reporting Summary linked to this paper.

## Data availability

All data generated or analyzed during this study are included in this published article and its supplementary information files. The data that support the findings of this study are available from the corresponding author upon reasonable request. Source data are provided with this paper.

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

## Acknowledgements

We are grateful to Dr. Nissim Hay for FoxO3a$^{-/-}$ cells. This work was supported by the National Institutes of Health RO1 grants EY029426, AI139768, EY024710 (to DS) and a NEI core grant (EY001792) and Illinois society for prevention of blindness. This content is solely the responsibility of the authors and does not necessarily represent the official views of the NIH.

## Author contributions

Conceptualization, R.K.S., and D.S.; Methodology, R.K.S., C.D.P., A.A., T.Y., J.M.A., L.K., and K.M.; Investigation, R.K.S., C.D.P., and D.S.; Writing original draft, R.K.S., R.K., and D.S.; Writing—Review and editing, R.K.S., R.K., L.M.S., and D.S.; Funding acquisition, R.K.S., D.S.; Supervision, D.S.

## Competing interests

The authors declare no competing interests.
