## [Peer Review File · Nature Communications]

mTORC2 confers neuroprotection and potentiates immunity during virus infectionREVIEWER COMMENTS

Reviewer #1 (Remarks to the Author):

As part of two distinct complexes, mTORC1 and mTORC2, the kinase mTOR has emerged as the central regulator of cell growth and proliferation by controlling most anabolic and catabolic processes in response to environmental cues. As such, mTOR is involved in fundamental physiological processes including immunity. Viral infection has been shown to activate, decrease and suppress mTOR signaling and different viruses have evolved mechanisms that target or co-opt mTOR pathways to promote virus replication, including HSV2 (PMID: 27231932). This virus-mTOR interaction is not only virus specific but also cell specific. mTORC1 is necessary for TLR-mediated production of type I IFN in innate immune cells. The role of mTORC2 in these processes is incompletely understood. However, in a mouse model of HSE mTORC2 was shown to be involved in TLR3-mediated type I IFN responses in astrocytes and neurones. A Nat Immunol publication by Sato et al. cited by the authors.

In this article, Suryawanshi et al. seek out to examine the role of mTORC2 in herpes simplex encephalitis. They make clever use of commercially available inducible model in which deletion of Rictor (a defining protein of the mTORC2 complex) can be conditionally induced upon treatment with tamoxifen.

The authors provide several pieces of new data, most notably the observation of decreased survival in *iRic*^{-/-} mice. The authors make some novel observations (see below detailed review), however there is a lack of methodological details and controls in several key experiments to adequately provide a mechanistic understanding about the role of mTORC2 in response to HSV1. In absence of mTORC2, the *in vivo* model that they present suggests increased susceptibility associated with increased virus load and spread into brain structures, leading to death. Although this is a compelling observation, the data presented do not inform about the role of Rictor-dependent mechanisms in these phenotypes.

Detailed review:

The authors use a model of corneal infection using a relatively high dose of virus (10E5). The type of infection (scarification or liquid) should be clarified as well as the virus strain used for the infection. The methods section names several HSV-1 strains. Different HSV-1 strains have different virulence in C57BL/6 mice providing helpful information to interpret the results. The clinical end-point whether weight or clinical signs should be clarified.

Regardless, Fig. 1 shows *iRic* mice to survive 60% by day 10 and 20% by day 12 post-infection compared to wild-type littermates (N should be indicated; males or females?). The decrease in survival is associated with increased virus load in corneal surfaces by day 4, and the trigeminal ganglia and brain stem by day 5 p.i.. However, the latter requires re-activation although for the brainstem is not entirely clear. Also, the choice of time-point to determine the viral load should be clarified. Is this the peak. This is of relevance for the interpretation of the immunophenotyping data presented in Fig. 2.

Fig. 2 examines the immune response of *iRic*-deficient mice. A) H&E stainings show decreased infiltration in *iRic*-deficient mice in a single micrograph is shown. A scale should be included. In addition, quantification of the data would provide stronger support to the conclusion. B-G) Several histograms show decreased % of several key immune populations. However data demonstrating the gating and details of the flow experiment should be provided to support the conclusion. It is not clear whether the decrease it is also observed before infection, uninfected controls should be shown. In the text there is a confusion between F and G (line 95: NK cells should be 2G). H) a single micrograph showing decreased anti-CD8 staining in *iRic*-deficient mice. I-J) serum neutralization assay showing

decreased humoral responses in *iRic*^{-/-} mice. This result is strong and it is consistent with previous observations of the role of mTORC2 in the development and function of Tfh cells, including humoral responses. K-J) A commercial chemokine/cytokine array show quite a range of variation in both sets of animals, however distinct patterns of expression can be discerned both in the cervical lymph nodes and the brain stem. The strong chemokine/cytokine response in wild-type mice at day 10 p.i. (as indicated in the figure legend) it is surprising. Is the virus active at this time point? Viral titer should be shown. The type I IFN response should be examined to clarify the phenotype of HSE susceptibility described in Fig1, and based on the extensive literature of the role of type I IFN in the ocular model. The experimental design should also consider earlier time-points to monitor immune responses which may precede dysregulation observed at day 10.

The authors switch gears to examine the activation of apoptosis in infected tissues and cells. Fig. 3 shows A) representative TUNNEL data from the mouse ocular system, (C-F) representative flow data from MEF cells and proteome. The authors claim increased apoptosis in HSV-1 infected *iRic*-deficient cells. However, a heatmap shown in Supplementary Fig. 3 shows relatively subtle differences between groups, and no discernable differences p53 abundance, between genotypes. (G) Z-VAD-FMK treatment decreases apoptosis in HSV1-infected MEF. The authors conclude that HSV-1 triggers apoptosis in *iRic*^{-/-}. However, to support the conclusions these data require both a negative control, i.e. *iRic*^{-/-}-MEF uninfected, to determine the basal level of apoptosis in *iRic*^{-/-} mice and whether this is increased upon infection— and a positive control of apoptosis such as staurosporin to determine if HSV-1 triggered apoptosis is specific (epatoside as used by the authors later). Also, considering the crucial HSV1 infection in neurons and astrocytes in HSE and the importance of cell-type specific host defense mechanisms, these studies would be stronger using *Rictor*^{-/-} brain cells.

An important additional control to link the data with the original finding of increased HSV-1 susceptibility in *iRic*^{-/-}-mice, it would be important to determine virus production. Increased apoptosis would be expected to lead to decreased virus production in *iRic*^{-/-} cells. However, the phenotype observed in vivo is decreased survival in the presence of increased viral load in *iRic*^{-/-} mice (Fig. 1). Although it requires further characterization, increased virus is also associated with some immune dysfunction (Fig. 2), including dysregulated secretion of inflammatory and regulatory cytokines and chemokines, which may also be contributing to the *iRic*^{-/-} phenotype of susceptibility to infection.

Fig. 3-I shows the presents original survival data showing increased survival of HSV1 infected *iRic*^{-/-} mice treated with the apoptosis inhibitor Z-VAD-FMK. This result is interesting however the conclusion of the authors that “loss of Rictor exacerbates HSV-1 apoptosis during infection that results in the death of the host in vivo” (line 177) is premature without further experimentation.

Fig. 4. The authors go on to investigate the Rictor-signaling pathway in corneal and neuronal cells as well as *iRic*^{-/-} MEF and wild-type cells. In wt MEF they observe phosphorylation of AKT, FOXO3a and FOXO1, which are absent in *iRic*. They provide suggestive confocal micrograph suggesting absence of FOXO3A in the cytoplasm of *iRic*^{-/-} infected cells. FOXO3a activates apoptosis downstream of AKT phosphorylation, hence *Foxo3a*^{-/-} cells show decreased apoptosis and increased virus protein and virus load. Supp Fig 5 shows representative micrograph of PI staining and the kinetics of PI staining in *iRic*^{-/-} cells either infected with HSV-1 or treated with etoposide, showing increased susceptibility to apoptosis in both conditions. The presence of HSV1 should be determined here, as done for *Foxo3a*^{-/-} cells (Fig 4).

Reviewer #2 (Remarks to the Author):

General comments

The authors showed that mTORC2 is required for surviving ocular HSV-1 infection. mTORC2 was required for mounting both innate and adaptive immune responses during HSV-1 infection. Furthermore, mTORC2 played a key role in protecting HSV-1 infected cells from apoptosis through AKT-dependent Foxo3a phosphorylation. These results suggest that mTORC2 organizes defense responses during HSE. Overall, the experiments were well designed and beautifully shown.

The reviewer has a concern about the role of apoptosis in HSV-1 infection. In Fig. 3, the authors, demonstrated that increased apoptosis exacerbated HSE in iRictor $-/-$ mice by showing the result that inhibition of apoptosis with Z-VAD increased the number of mice having survived HSV-1 infection (Fig. 3i). In Fig. 4, FoxO3a $-/-$ cells were resistant to cell death during HSV-1 infection (Fig. 4k). The reviewer expected that decreased cell death would decrease virus infection. However, higher virus yield in FoxO3a $-/-$ cells suggests that cell death is required for the control of virus infection (Fig. 4f, 4h). How these results can be consistent with each other? The authors need to show virus yields in infected iRictor $-/-$ MEFs, and explain the roles of cell death in the HSE model used in the present study.

Another concern is on the relationship between the first half and the latter half of this manuscript. Although mTORC2 is required for controlling both immune responses and cell death, the relationship between these two responses remains unclarified. The authors need to study the relationship by examining immune responses to HSV-1 in Z-VAD-treated mice.

Minor point

1. In Fig. 4e, HSV-1(GFP-tagged) is not visible. Why?
2. Fig. 2f and 2g are mistakenly placed.
3. In Fig.2 k&l, Swap $-/-$ and $+/+$ to make the results easier to understand.
4. In Fig.3 f, the authors showed that the information of apoptotic array. The authors should show the results in Supplementary data 3 in Main figure to make the results easier to understand.
5. In Fig. 4c, replace figure of p-Foxo3a to clarify activation of Foxo3a.
6. In Supplementary Fig. 1c, replace figure of Rictor to clarify efficiency of KO.

Reviewer #1 (Remarks to the Author):

As part of two distinct complexes, mTORC1 and mTORC2, the kinase mTOR has emerged as the central regulator of cell growth and proliferation by controlling most anabolic and catabolic processes in response to environmental cues. As such, mTOR is involved in fundamental physiological processes including immunity. Viral infection has been shown to activate, decrease and suppress mTOR signaling and different viruses have evolved mechanisms that target or co-opt mTOR pathways to promote virus replication, including HSV2 (PMID: 27231932). This virus-mTOR interaction is not only virus specific but also cell specific. mTORC1 is necessary for TLR-mediated production of type I IFN in innate immune cells. The role of mTORC2 in these processes is incompletely understood. However, in a mouse model of HSE mTORC2 was shown to be involved in TLR3-mediated type I IFN responses in astrocytes and neurons. A Nat Immunol publication by Sato et al. cited by the authors.

In this article, Suryawanshi et al. seek out to examine the role of mTORC2 in herpes simplex encephalitis. They make clever use of commercially available inducible model in which deletion of Rictor (a defining protein of the mTORC2 complex) can be conditionally induced upon treatment with tamoxifen.

The authors provide several pieces of new data, most notably the observation of decreased survival in iRic^{-/-} mice. The authors make some novel observations (see below detailed review), however there is a lack of methodological details and controls in several key experiments to adequately provide a mechanistic understanding about the role of mTORC2 in response to HSV1. In absence of mTORC2, the in vivo model that they present suggests increased susceptibility associated with increased virus load and spread into brain structures, leading to death. Although this is a compelling observation, the data presented do not inform about the role of Rictor-dependent mechanisms in these phenotypes.

Detailed review:

The authors use a model of corneal infection using a relatively high dose of virus (10E5). The type of infection (scarification or liquid) should be clarified as well as the virus strain used for the infection.

Answer: The infection was performed after corneal scarification and HSV-1 (McKrae) was used for all animal infections. The details have now been added in the results and methods section.

The methods section names several HSV-1 strains. Different HSV-1 strains have different virulence in C57BL/6 mice providing helpful information to interpret the results.

Answer: For our animal studies we used HSV-1 McKrae, which is a neurovirulent strain. This is now clarified in the text (Line 358-359)

The clinical end-point whether weight or clinical signs should be clarified.
Answer: The clinical endpoints included weight loss of greater 20% and hunched posture of the animal. The details have now been added to the methods section. (Line 360-361)

Regardless, Fig. 1 shows iRic mice to survive 60% by day 10 and 20% by day 12 post-infection compared to wild-type littermates (N should be indicated; males or females?).
Answer: Female mice were used in all animal experiments; respective details have now been added to the methods section. For the survival curve in the figure 1 a group of 10 animals was used. The information about n value have now been updated in all the experiments.

The decrease in survival is associated with increased virus load in corneal surfaces by day 4, and the trigeminal ganglia and brain stem by day 5 p.i.. However, the latter requires re-activation although for the brainstem is not entirely clear.

Answer: We clearly detected virus in the tissues. Since HSV-1 infection establishes latency in the trigeminal ganglion, the re-activation experiment was performed to compare the amount of latent virus present in the iRic -/- verses iRic +/+, the explanation for re-activation experiment has been added in the results section Line 69-71). Similarly, active virus replication in the trigeminal ganglion is now shown in Fig 1h. and the brain stem showing active virus infection is shown in fig 1 j and k.

Also, the choice of time-point to determine the viral load should be clarified. Is this the peak. This is of relevance for the interpretation of the immunophenotyping data presented in Fig. 2.

Answer: The day 4 post infection showed statistically significant differences in the virus shedding at ocular surface when infected with 1×10^5 . However, with higher MOI (5×10^5) we have seen statistically significant differences at day 2 post infection as well. The peak of virus shedding appears on day 2 post infection, now explained in the text (Line number 61-62). The day 4 post infection differences show relevance with the TUNNEL staining in fig 2. We have now interpreted it in the text.

Fig. 2 examines the immune response of iRic-deficient mice. A) H&E staining's show decreased infiltration in iRic-deficient mice in a single micrograph is shown. A scale should be included. In addition, quantification of the data would provide stronger support to the conclusion.

Answer: As per suggestion the H & E staining images have been quantified for infiltrated cells and the corneal thickness representing inflammation of HSV-1 infected cornea (Figure 2a-b).

B-G) Several histograms show decreased % of several key immune populations. However, data demonstrating the gating and details of the flow experiment should be provided to support the conclusion.

Answer: The gating strategy and flow experiment details have now been provided in supplementary and methods section.

It is not clear whether the decrease it is also observed before infection, uninfected controls should be shown.

Answer: As per suggestion the account of immune cell population was taken for mock infected animals and the data is now added as a new supplementary data 2 and discussed in the text (line 99-101).

In the text there is a confusion between F and G (line 95: NK cells should be 2G).

Answer: Thank you very much for the careful observation, we have now made the correction in the text. Line number 96 and 97.

I-J) serum neutralization assay showing decreased humoral responses in iRic^{-/-} mice. This result is strong, and it is consistent with previous observations of the role of mTORC2 in the development and function of Tfh cells, including humoral responses.

Answer: Thank you for highlighting the importance of the data.

K-J) A commercial chemokine/cytokine array show quite a range of variation in both sets of animals, however distinct patterns of expression can be discerned both in the cervical lymph nodes and the brain stem.

Answer: The distinct pattern of cytokine expression may lead to importance of mTORC2 in homeostasis of anti-inflammatory cytokines.

The strong chemokine/cytokine response in wild-type mice at day 10 p.i. (as indicated in the figure legend) it is surprising. Is the virus active at this time point? Viral titer should be shown.

Answer: We did not observe the presence mature virus particle at day 10 post infection however the cytokine response was observed at day 10 post infection and may be correlated with the primary immune response.

The type I IFN response should be examined to clarify the phenotype of HSE susceptibility described in Fig1, and based on the extensive literature of the role of type I IFN in the ocular model. The experimental design should also consider earlier time-points to monitor immune responses which may precede dysregulation observed at day 10.

Answer: Following up on this important suggestion we performed an in-vitro experiment with iRic^{+/+} and iRic^{-/-} cells infected with HSV-1 or mock. cytokine expression involved in IFN I response was analyzed as demonstrated in Supplementary data 7 and the data have been discussed in the text.

The authors switch gears to examine the activation of apoptosis in infected tissues and cells. Fig. 3 shows A) representative TUNNEL data from the mouse ocular system, (C-F) representative flow data form MEF cells and proteome. The authors claim increased apoptosis in HSV-1 infected iRic-deficient cells. However, a heatmap shown in Supplementary Fig. 3 shows relatively subtle differences between groups, and no discernable differences p53 abundance, between genotypes.

Answer: The heatmap have now been modified for color intensity and according to another reviewers' suggestion have been moved to main figure 3.

We agree that there are subtle differences between groups but it may be attributed to single time point (12hpi) selected for apoptosis array. It has been reported that a modest change in the dynamic balance of BCL-2 proteins is known to inhibit apoptosis (Fujise et al., 2000; You et al., 2017),

However, we observed anti-apoptotic proteins including heat shock protein (HSP-70) and myeloid leukemia cell differentiation protein 1 (MCL-1) were preferentially upregulated in infected RICTOR+/+ cells whereas several pro-apoptotic proteins including, tumor necrosis factor receptor 1 (TNFR1), hypoxia inducing factor (HIF-1 α) and HSP-27 were upregulated in infected RICTOR-/- cells. Increased HIF1- α may generate hypoxic conditions responsible for the rise in pro-apoptotic proteins (Carmeliet et al., 1998; Greijer and van der Wall, 2004). Sustained activation of AKT increases MCL-1 (Kuo et al., 2001; Longo et al., 2008). mTORC2 is known to stabilize MCL-1 protein (Koo et al., 2015), which explains lower levels of MCL-1 in RICTOR-/- cells.

References:

- Fujise, K., Zhang, D., Liu, J., and Yeh, E.T. (2000). Regulation of apoptosis and cell cycle progression by MCL1. differential role of proliferating cell nuclear antigen. *J. Biol. Chem.* 275, 39458-39465.
- You, Y., Cheng, A., Wang, M., Jia, R., Sun, K., Yang, Q., Wu, Y., Zhu, D., Chen, S., Liu, M., et al. (2017). The suppression of apoptosis by α -herpesvirus. *Cell Death Dis.* 8, e2749.
- Carmeliet, P., Dor, Y., Herbert, J.M., Fukumura, D., Brusselmans, K., Dewerchin, M., Neeman, M., Bono, F., Abramovitch, R., Maxwell, P., et al. (1998). Role of HIF-1 α in hypoxia-mediated apoptosis, cell proliferation and tumour angiogenesis. *Nature.* 394, 485-490.
- Greijer, A.E., and van der Wall, E. (2004). The role of hypoxia inducible factor 1 (HIF-1) in hypoxia induced apoptosis. *J. Clin. Pathol.* 57, 1009-1014.
- Kuo, M.L., Chuang, S.E., Lin, M.T., and Yang, S.Y. (2001). The involvement of PI 3-K/akt-dependent up-regulation of mcl-1 in the prevention of apoptosis of Hep3B cells by interleukin-6. *Oncogene.* 20, 677-685.
- Longo, P.G., Laurenti, L., Gobessi, S., Sica, S., Leone, G., and Efremov, D.G. (2008). The akt/mcl-1 pathway plays a prominent role in mediating antiapoptotic signals downstream of the B-cell receptor in chronic lymphocytic leukemia B cells. *Blood.* 111, 846-855.
- Koo, J., Yue, P., Deng, X., Khuri, F.R., and Sun, S. (2015). mTOR complex 2 stabilizes mcl-1 protein by suppressing its glycogen synthase kinase 3-dependent and SCF-FBXW7-mediated degradation. *Mol. Cell. Biol.* 35, 2344-2355.

(G) Z-VAD-FMK treatment decreases apoptosis in HSV1-infected MEF. The authors conclude that HSV-1 triggers apoptosis in iRic-/- . However, to support the conclusions these data require both a negative control, i.e. iRic-/- MEF uninfected, to determine the basal level of apoptosis in iRic-/- mice and whether this is increased upon infection– and a positive control of apoptosis such as staurosporin to determine if HSV-1 triggered apoptosis is specific (etoposide as used by the authors later).

Answer: As per suggestion the experiment has been performed and the data has been added to Supplementary data 4e and f. As per the results basal level of apoptosis is more in iRic-/- cells and it increases with etoposide treatment or infection. Line number 150-153.

Also, considering the crucial HSV1 infection in neurons and astrocytes in HSE and the importance of cell-type specific host defense mechanisms, these studies would be stronger using Rictor-/- brain cells.

Answer: As per the suggestion we tried to culture the neurons from WT and iRic -/- animals. Similarly, the treatment of neurons isolated from iRic-/- cells were not able to

tolerate the treatment of 4-hydroxytamoxifen and unable to survive for 72 hours which is a time required to knockout the rictor. This might be happening because of higher base level apoptosis also observed in iRic^{-/-} MEF cells now presented in supplementary data 4e and f. However, we were successful in culturing the neurons from WT animals and using these neurons we studied the AKT-FoxO3a axis and is explained in Figure 4c. The results for neurons are in line with human corneal epithelial cell presented in Figure 4b, which shows phosphorylation of AKT and FoxO3a is a key step during HSV-1 infection.

An important additional control to link the data with the original finding of increased HSV-1 susceptibility in iRic^{-/-} mice, it would be important to determine virus production. Increased apoptosis would be expected to lead to decreased virus production in iRic^{-/-} cells. However, the phenotype observed in vivo is decreased survival in the presence of increased viral load in iRic^{-/-} mice (Fig. 1). Although it requires further characterization, increased virus is also associated with some immune dysfunction (Fig. 2), including dysregulated secretion of inflammatory and regulatory cytokines and chemokines, which may also be contributing to the iRic^{-/-} phenotype of susceptibility to infection.

Answer: Thank you very much for this important suggestion. We performed the suggested experiment and the data have been presented in a new supplementary figure number 6. The results have now been discussed in the text (line number 192-202).

Fig. 3-I shows the original survival data showing increased survival of HSV1 infected iRic^{-/-} mice treated with the apoptosis inhibitor Z-VAD-FMK. This result is interesting however the conclusion of the authors that “loss of Rictor exacerbates HSV-1 apoptosis during infection that results in the death of the host in vivo” (line 177) is premature without further experimentation.

Answer: The concluding statement has been modified to “loss of Rictor may be responsible to induce host apoptotic response during infection”.

Fig. 4. The authors go on to investigate the Rictor-signaling pathway in corneal and neuronal cells as well as iRic^{-/-} MEF and wild-type cells. In wt MEF they observe phosphorylation of AKT, FOXO3a and FOXO1, which are absent in iRic. They provide suggestive confocal micrograph suggesting absence of FOXO3A in the cytoplasm of iRic^{-/-} infected cells. FOXO3a activates apoptosis downstream of AKT phosphorylation, hence Foxo3a^{-/-} cells show decreased apoptosis and increased virus protein and virus load. Supp Fig 5 shows representative micrograph of PI staining and the kinetics of PI staining in iRic^{-/-} cells either infected with HSV-1 or treated with etoposide, showing increased susceptibility to apoptosis in both conditions. The presence of HSV1 should be determined here, as done for Foxo3a^{-/-} cells (Fig 4).

Answer: As per suggestion we have now added the data of virus infection in iRic^{-/-} cells (Supplementary data 6 a-d) and as expected we see the restriction of virus replication in iRic^{-/-} cells which is now discussed in the text (line 192-202).

Reviewer #2 (Remarks to the Author):

General comments

The authors showed that mTORC2 is required for surviving ocular HSV-1 infection. mTORC2 was required for mounting both innate and adaptive immune responses during HSV-1 infection. Furthermore, mTORC2 played a key role in protecting HSV-1 infected cells from apoptosis through AKT-dependent Foxo3a phosphorylation. These results suggest that mTORC2 organizes defense responses during HSE. Overall, the experiments were well designed and beautifully shown.

The reviewer has a concern about the role of apoptosis in HSV-1 infection. In Fig. 3, the authors, demonstrated that increased apoptosis exacerbated HSE in iRictor $-/-$ mice by showing the result that inhibition of apoptosis with Z-VAD increased the number of mice having survived HSV-1 infection (Fig. 3i). In Fig. 4, FoxO3a $-/-$ cells were resistant to cell death during HSV-1 infection (Fig. 4k). The reviewer expected that decreased cell death would decrease virus infection. However, higher virus yield in FoxO3a $-/-$ cells suggests that cell death is required for the control of virus infection (Fig. 4f, 4h). How these results can be consistent with each other? The authors need to show virus yields in infected iRictor $-/-$ MEFs, and explain the roles of cell death in the HSE model used in the present study.

Answer: Thank you for this important suggestion, We have now added the data of virus infection in iRic $-/-$ cells (Supplementary data 6) and as expected we see the restriction of virus replication in iRic $-/-$ cells which is now discussed in the text (line 192-202). In contrast in-vivo we observe increased virus replication in iRic $-/-$ mice which is associated with some immune dysfunction (Fig. 2), including dysregulated secretion of inflammatory and regulatory cytokines.

Another concern is on the relationship between the first half and the latter half of this manuscript. Although mTORC2 is required for controlling both immune responses and cell death, the relationship between these two responses remains unclarified. The authors need to study the relationship by examining immune responses to HSV-1 in Z-VAD-treated mice.

Answer: As per the suggestion we performed an experiment to evaluate the immune response of HSV-1 infected iRic $-/-$ mice treated with either mock or Z-VAD. The results of the experiments have been incorporated as Supplementary data 5 and have been discussed in the text (line number 186-191).

Minor point

1. In Fig. 4e, HSV-1(GFP-tagged) is not visible. Why?

Answer: Actually iRic $-/-$ cells grow very less virus, because in the absence of rictor the virus infected cells undergo apoptosis which limits spread of virus. These results have now been explained with the new supplementary data 6. However, in order to visualize the intracellular location of FoxO3a we searched for virus infected cells in iRic $-/-$ population and small dots of GFP were observed around the infected cells (because of

very less replication of virus), which also showed intranuclear localization of FoxO3a (Figure 4e). Because of this reason HSV-1 in iRic $-/-$ cells is not clearly visible.

2. Fig. 2f and 2g are mistakenly placed.

Answer: Thank you for careful observation. The change has been made in the text. (Line 96)

3. In Fig.2 k&l, Swap $-/-$ and $+/+$ to make the results easier to understand.

Answer: The Fig.2 k and l have now been swapped for easier understanding in line with other figures.

4. In Fig.3 f, the authors showed that the information of apoptotic array. The authors should show the results in Supplementary data 3 in Main figure to make the results easier to understand.

Answer: As per suggestion the figure has now been moved to the main figure 3g.

5. In Fig. 4c, replace figure of p-Foxo3a to clarify activation of Foxo3a.

Answer: The figure has been replaced for clarification.

6. In Supplementary Fig. 1c, replace figure of Rictor to clarify efficiency of KO.

Answer: The background contrast in the figure has now changed to clarify the efficiency of rictor knockout.

REVIEWER COMMENTS

Reviewer #1 (Remarks to the Author):

The authors have adequately addressed this reviewer's comments.

The results support the conclusion for the role of Rictor, an essential component of mTORC2, in neuroprotection and host response to protect from lethal HSV-1 infection.

Reviewer #2 (Remarks to the Author):

The comments for us is enough to clear up our concern. I think the revised version is acceptable for publication.